# Gender Specificities of Cardiac Troponin Serum Levels: From Formation Mechanisms to the Diagnostic Role in Case of Acute Coronary Syndrome

**DOI:** 10.3390/life13020267

**Published:** 2023-01-18

**Authors:** Aleksey Michailovich Chaulin

**Affiliations:** 1Department of Histology and Embryology, Samara State Medical University, Samara 443099, Russia; a.m.chaulin@samsmu.ru or alekseymichailovich22976@gmail.com; Tel.: +7-(927)-770-25-87; 2Department of Cardiology and Cardiovascular Surgery, Samara State Medical University, Samara 443099, Russia

**Keywords:** cardiac troponins, troponin T, troponin I, gender specificities, mechanisms, diagnostic role, acute coronary syndrome

## Abstract

Cardiac troponins T and I are the main (most sensitive and specific) laboratory indicators of myocardial cell damage. A combination of laboratory signs of myocardial cell damage (elevated levels of cardiac troponins T and I) with clinical (severe chest pain spreading to the left side of the human body) and functional (rise or depression of the ST segment, negative T wave or emergence of the Q wave according to electrocardiography and/or decrease in the contractility of myocardial areas exposed to ischemia according to echocardiography) signs of myocardial ischemia is indicative of the ischemic damage to cardiomyocytes, which is characteristic of the development of acute coronary syndrome (ACS). Today, with early diagnostic algorithms for ACS, doctors rely on the threshold levels of cardiac troponins (99th percentile) and on the dynamic changes in the serum levels over several hours (one, two, or three) from the moment of admission to the emergency department. That said, some recently approved highly sensitive methods for determining troponins T and I show variations in 99th percentile reference levels, depending on gender. To date, there are conflicting data on the role of gender specificities in the serum levels of cardiac troponins T and I in the diagnostics of ACS, and the specific mechanisms for the formation of gender differences in the serum levels of cardiac troponins T and I are unknown. The purpose of this article is to analyze the role of gender specificities in cardiac troponins T and I in the diagnostics of ACS, and to suggest the most likely mechanisms for the formation of differences in the serum levels of cardiac troponins in men and women.

## 1. Introduction

Atherosclerotic lesions of coronary vessels are the morphological substrate of the main clinical forms of coronary heart disease which, depending on a number of conditions (lifestyle, use of medicines, presence of concomitant diseases and other factors), may be characterized by stable progression (stable angina pectoris) and by the advance of the pathological process, manifested in the form of exacerbations, which are commonly referred to as “acute coronary syndrome” (ACS). The main symptom of ACS is pain or discomfort in the chest, which forces the patient to call an ambulance or visit the emergency cardiology department on his/her own [1,2,3]. From the point of view of a clinician, ACS is a temporary (“provisional”) diagnosis that is necessary for the initial assessment of the current situation, risk stratification, and the choice of treatment strategy for patients with an exacerbation of coronary heart disease [4]. After additional diagnostic manipulations (electrocardiography, echocardiography, laboratory tests, coronary angiography), the diagnosis of ACS “transforms” into the following clinical forms: unstable angina, myocardial infarction without ST elevation, myocardial infarction with ST segment elevation, or non-coronarogenic (non-cardiac) disease (Figure 1). In the latter case, the pain may be caused not by atherosclerosis of the coronary vessels, but by the intercostal neuralgia, or pain, spreading due to the pathology of the abdominal organs (pancreatitis, gastritis, etc.) or the musculoskeletal system (osteochondrosis).

According to the statistics of the World Health Organization, ACS is one of the most dangerous forms of cardiovascular pathology and occupies a leading place in the structure of mortality in most countries in the world [5]. According to the Eurasian Association of Cardiologists, the highest mortality rate in patients from ACS among European countries is recorded in Russia, Ukraine, Belarus, Bulgaria, and Lithuania. Among European countries, the hospital mortality rate ranges from 6 to 14%, while in Russia the mortality rate was 18.6% and 17.7% in 2015 and 2016, respectively [4,6]. Prevalence of ACS is lower among women than among men. Thus, the proportion of women among patients with ACS, according to the Russian RECORD-3 register for 2015, was 39%. Among patients with myocardial infarction with and without ST elevation, women also constituted a minority—32% and 44%, respectively [7].

Since chest pain may be the first clinical sign of ACS/myocardial infarction, its early determination is associated with improved clinical outcomes. At the same time, quick exclusion of the diagnosis of myocardial infarction can reduce costly hospitalizations and prevent unnecessary, expensive, and potentially dangerous procedures [8].

Current Russian [8,9] and international guidelines [5,10] recommend the use of cardiac troponin tests as the “gold standard” for diagnosing myocardial infarction. This is due to their high cardiospecificity (localization only in myocardial cells) [11,12], diagnostic and prognostic value in forecasting myocardial infarction in case of acute conditions, and the risk of all-cause mortality and cardiovascular events in the general population [13,14].

Due to differences in the prevalence of ACS [7,15] and in the degree of increase in cardiac troponin levels in men and women at the early stages of diagnostics, several authors propose a gender-based approach to the early diagnostics of ACS [16]. A gender-oriented approach to early diagnosis of ACS implies the use of different threshold levels (99th percentile) and kinetics in increased cardiac troponins in men and women [16,17]. However, to date, this approach is not sufficiently covered in a number of guidelines and there are no specific recommendations for the diagnosis of ACS depending on gender identity. This is largely due to the inconsistency of the results of clinical studies performed in this regard. In addition, the specific physiological mechanisms underlying the formation of gender-based variations in serum levels of cardiac troponins have not been established.

The purpose of this review is to systematize information on the importance of taking into account the data on the gender (sex) characteristics of the content (in the range of the 99th percentile) of cardiac troponins in the diagnostics and prognosis of the development of ACS, as well as on the possible mechanisms for the formation of gender differences in the cardiac troponin content levels.

## 2. Characteristics and Benefits of Using New Highly Sensitive Laboratory Tests

The cardiac troponin complex regulates striated muscle contraction and consists of three subunits: troponin C, T, and I, which are designated according to their functional significance. Troponin C (the calcium-binding subunit) binds to calcium ions, which initiates conformational changes in the troponin complex and tropomyosin, leading to the opening of myosin-binding sites on the actin molecule. Subsequently, the myosin head interacts with the myosin-binding sites, resulting in the formation of transverse (actin-myosin) bridges. Troponin T binds to tropomyosin, attaching the troponin complex to the thin (actin) sarcomere filaments. Troponin I binds to actin and reduces the affinity of troponin C for calcium, thereby inhibiting actin-myosin interactions [18,19].

The majority of the cardiac troponin (approximately 95%) is associated with myofilaments, and a small concentration is in a free state in the cytosol. After the loss of the membrane integrity of cardiac cardiomyocytes, troponin is initially released from the cytosol into the cardiac interstitium and then into the peripheral blood. Troponins T and I are present in cardiac and skeletal muscles and are encoded by different genes in each muscle type, resulting in two immunologically different products. Laboratory diagnostic studies are based on the use of high-affinity antibodies that are specific for cardiac troponin T and cardiac troponin I, and not for troponin C, since it is identical in both muscles [19,20,21].

Since H. Katus et al. described the first test to measure cardiac troponin in 1991 [22], an extremely long journey has been made from the development to the introduction of highly sensitive troponin tests into clinical practice and the emergency care department [23,24].

The use of highly sensitive troponin laboratory diagnostic tests is an important step forward due to their high sensitivity to cardiomyocyte necrosis, as they are able to determine cardiac troponin concentrations approximately 10–100 times lower than conventional tests, leading to more accurate and timely diagnoses [25,26].

According to the International Federation of Clinical Chemistry (IFCC) guidelines, two criteria are used to define this new generation of (highly sensitive) troponin tests: (1) the coefficient of variation at the 99th percentile value should be 10% or less (the most optimal immunoassays), although tests with an error of more than 10, and 20% or less, are still considered clinically acceptable; (2) the concentration of cardiac troponins should be above the minimum determinable concentration (determination limit) in more than 50% of healthy individuals [23,27,28].

New laboratory diagnostic tests allow the implementation of earlier diagnostics and the quicker exclusion of myocardial infarction due to high sensitivity [29]; however, the second key immunoassay criterion, specificity, is affected significantly. Clinically, this is expressed by the presence of a wide range of other troponin-positive non-cardiac and cardiac conditions other than myocardial infarction [30,31]. Although not all mechanisms of the elevation of troponin levels are known, in some conditions, this may be due to a decrease in oxygen supply to the myocardium. It is not clear whether the damage is always irreversible, which inevitably leads to myocardial necrosis, or whether some diseases can cause reversible damage. The main conditions (physiological and pathological) that cause an increase in the levels of cardiac troponins are shown in Table 1.

Currently, all available highly sensitive methods for determining cardiac troponins have a single diagnostic threshold value for ACS diagnostics, based on the value of the 99th percentile, which is calculated for a healthy population [28,32]. However, this threshold value can vary significantly depending on the methodology for determining (manufacturer) cardiac troponins [32,33]. According to the IFCC, the main manufacturers of highly sensitive immunological reagent kits for determining cardiac troponins are: Abbot (Chicago, IL, USA), Beckman Coulter (Brea, CA, USA), bioMerieux (Marcy-l’Étoile, France), ET Healthcare (Shanghai, China), LSI Medience (Tokyo, Japan), Fujirebio (Tokyo, Japan), Ortho Clinical Diagnostics (Raritan, NJ, USA), Quidel/Alere (San Diego, CA, USA), Roche (Basel, Switzerland), Siemens (Munich, Germany), etc. (Table 2).

The additional important diagnostic advantage of highly sensitive tests for the immunological determination of cardiac troponins is the ability to determine myocardial damage at the subclinical level, which can be used to monitor and assess the prognosis of patients suffering from a number of chronic pathologies, including coronary heart disease [34], and during the treatment of oncological diseases by using chemotherapeutic compounds, which are characterized by cardiotoxicity [35,36], as well as chronic obstructive pulmonary disease [37], chronic kidney disease [38], diabetes mellitus [39], arterial hypertension [40], etc.

An important advantage of modern highly sensitive troponin immunotests is the ability to detect cardiac troponin molecules in noninvasively obtained biological material (oral fluid, urine, sweat) [41,42,43,44,45,46,47,48]. When receiving these biomaterials, a number of advantages can be noted: the process is non-invasive, atraumatic, and painless, no specially trained medical personnel is required, and the preliminary diagnostics can be tested at home (test strips). Therefore, this method will allow for the diagnosis of diseases in a non-invasive way. However, to date, this is a little-studied and controversial area that requires further research to confirm these possibilities.

## 3. Gender Specificities of Cardiac Troponin Levels

Gender specificities in concentrations are characteristic of a number of laboratory analytes (RBC count, hemoglobin, creatinine concentrations, etc.) which are widely used in modern clinical practice. As for cardiac markers, for the first time, information about the gender specificities of laboratory tests was found during the study of creatine kinase (CK) activity, the test for determining which was used to diagnose myocardial infarction in the 1960–1970s of the XX century. Healthy men had significantly higher CK activity than women, and dark-skinned people had higher CK activity than Caucasians [49]. This was also characteristic of the MB-fraction of creatine kinase (CK–MB), both for activity (u/l) and for the concentration of CK–MB (CK–MB–mass), measured in ng/mL. The mechanism of these differences, according to academic specialists, was largely due to the differences in skeletal muscle mass in men compared to women [49]. Later, gender differences were noted for natriuretic peptides, and, according to the authors, they were due to the different influence of male and female sex hormones on the production of natriuretic peptides in the myocardium [50]. However, with the introduction of the first tests for immunological determination of cardiac troponins, gender features ceased to be determined, which was probably due to the low sensitivity of these test systems, because they determined troponin concentrations in only 5% of healthy subjects [24,25,26]. Therefore, at that time, the prevailing opinion was that cardiac troponins are strictly intracellular molecules that appear in the blood serum only in the case of serious pathologies of the myocardium, and certain positive levels of troponins in patients with unconfirmed myocardial infarction were most often interpreted as false positive results. This opinion was also supported by the studies reporting a high prevalence of false positive results of cardiac troponins in patients with rhabdomyolysis in cases of skeletal muscle pathologies [51,52]. As the sensitivity of laboratory methods increased, cardiac troponin levels began to be determined in the blood of a larger number of healthy individuals (which allowed cardiac troponins to be regarded as the metabolic products of cardiomyocytes) and the first reports of possible gender-based variations in cardiac troponin levels appeared. Thus, F. Apple et al., studying the reference limits of cardiac troponin levels in a large sample of patients (*n* = 686) using eight immunological determination tests, found the presence of gender variations in two methods of immunological determination of cardiac troponin I. At the same time, the average level of troponin I in men was 1.2–2.5 times higher than in women, according to the nonparametric statistical analysis of the results [53]. However, this study is actually the only one that reported any gender variations for the moderately sensitive research methods, and therefore this was not reflected in practical medicine. With the introduction of highly sensitive immunological tests, it has been shown that in 80% of healthy individuals, determinable cardiac troponin concentrations exceed the determination limit [54], and the rates are significantly higher in men than in women, leading to a more detailed study of the potential gender-specific 99th percentile. In a large study including 524 healthy subjects (272 males, 252 females), 99th percentile levels were calculated for 19 troponin tests: 1 cardiac troponin T test by Roche, and 18 troponin I tests by Abbott, Alere, Beckman, bioMerieux, Instrumentation Laboratory, Ortho–Clinical Diagnostics, Singulex, Siemens and Roche, of which five were analytically classified as highly sensitive. The study found that 99th percentile levels exceeded the determination limit in 80% of people in the case of highly sensitive immunological determination tests, while moderately sensitive tests determined measurable troponin levels in about 25% to 30% of patients. Gender specificities of the 99th percentile were typical of all highly sensitive test systems for determining cardiac troponin I, the values of which were 1.2–2.4 times higher in men than in women. Approximately similar values were demonstrated by the highly sensitive analysis for cardiac troponin T: the 99th percentile for men was 20 ng/L, and for women it was 13 ng/L, while the overall (regardless of gender) calculated 99th percentile was 15 ng/L. The gender-specific 99th percentile was also characteristic of some moderately sensitive test systems, according to which troponin levels were 1.3–5 times higher in men than in women [54].

A. K. Saenger et al. [55] showed that statistically significant differences were observed in high-sensitivity troponin T concentrations in men and women, with the 99th percentile limit for men (15.5 ng/L) approximately 1.7 times higher than for women (9.0 ng/L).

In another large study, M. Gore et al. reported similar results in the three large independent patient cohorts in which highly sensitive troponin T concentrations were analyzed based on age, sex, and race stratifications. It is important to note that more than 10% of men aged 65 to 74 years without cardiovascular disease had high-sensitivity troponin T values above the threshold value (99th percentile) (>14 ng/L). In each cohort studied, the value of the 99th percentile increased with age over 60 and was higher in men than in women. The academic specialists also found significant differences in the threshold levels of highly sensitive troponin T depending on age and gender specificities: men (disregarding age) = 23 ng/L, men 50–64 years old = 28 ng/mL, men under 50 years old = 19 ng/L l; women (disregarding age) = 9 ng/mL, women 50–64 years old = 14 ng/mL, women under 50 years old = 9 ng/mL [56]. Thus, gender and age must be taken into account when calculating the 99th percentile levels of highly sensitive troponins, whereas the use of the single threshold value (14 ng/L) for highly sensitive troponin T analysis can lead to overdiagnosis of myocardial infarction, especially in men and the elderly, since their normal (baseline) level significantly exceeds the 99th percentile recommended by the immunotests manufacturer. The studies presented are indicative of the need for a close study of the gender-age specificities of troponins for clinical validation.

In the current European Society of Cardiology (ESC) guidelines for the diagnostics and treatment of myocardial infarction without ST-segment elevation [57], the diagnosis of myocardial infarction is based not on the single value of the cardiac troponin, but on two main algorithms based on the dynamic changes in cTn at the zero moment (on admission to the emergency care department and first blood test) and after 3 h or after 1 h. Only validated, highly sensitive troponin immunotests with confirmed threshold levels or cutoff values should be used to apply these algorithms. Notably, the 0/3 h algorithm makes a clear reference to the upper control limit of the 99th percentile, and this is also specified in the fourth universal definition of myocardial infarction [10], while the 0/1 h algorithm uses cutoffs below the 99th percentile, calculated for specific troponin tests of immunological determination. The most important role in these diagnostic algorithms is played by the kinetics of the increase in cardiac troponin levels during the first hours from the moment of chest pain/admission to the emergency care department. The positive predictive value of these algorithms for patients with myocardial infarction, i.e., those who meet the “rule-in” criteria, is 75–80%. Some patients that meet the “rule-in” criteria with diagnoses other than myocardial infarction may have conditions (e.g., takotsubo cardiomyopathy, myocarditis, etc.) that usually require hospitalization and coronary angiography for accurate diagnosis [57]. As the upper control limit of the 99th percentile is not always gender-specific, and the 0/1 h algorithm does not use gender-based cutoffs, the absence of specificity and the relatively low positive predictive value of cardiac troponins in patients with myocardial infarction may be partly explained by the inadequate threshold value, which is equal for both men and women.

There is ongoing debate regarding the appropriateness of using the gender-specific 99th percentile as the diagnostic threshold [26,57,58,59,60,61]. Its use can lead to an excess of patients with the elevated cardiac troponins level which is not associated with myocardial infarction [58,59]. On the other hand, the use of common cutoffs may lead to underestimation of myocardial infarction, especially in women [60,61]. V. Novack et al. [62] showed that women form a high-risk group which receives less of the treatment methods for myocardial infarction than recommended by the guidelines, including less frequent cardiac catheterization and use of secondary prevention methods. That is why determination of the threshold level of cardiac troponin in women is critical, since an incorrect decision limit can lead to incorrect interpretation of the result and further mismanagement of these patients.

In a retrospective study by C. Trambas et al. [63], switching from a moderately sensitive method for determining troponin I to a highly sensitive troponin I test significantly increased the number of patients with elevated troponin I concentrations, while no statistically significant changes were found in men. The introduction of gender-specific threshold reference values did not lead to an increase in the number of cases of myocardial infarction among the female population. On the other hand, the introduction of gender-specific reference intervals has identified those women who are at increased risk of future cardiovascular events. Similar results were demonstrated in cases of the use of high-sensitivity troponin T in the study (a study of bypass angioplasty revascularization in cases of type 2 diabetes) [64]. Within this study, they observed 684 women and 1601 men with type 2 diabetes mellitus and stable coronary artery disease for 5 years. The results showed that among patients with type 2 diabetes mellitus and stable coronary artery disease, women with circulating levels of high-sensitivity troponin T that are within the “normal” range (the commonly used 99th percentile disregarding gender) are at increased risk of serious cardiovascular events, which exceeds the rates observed among men with similar concentrations of highly sensitive troponin T [64]. Thus, this study also shows the need to revise the 99th percentile with account taken of gender.

## 4. Possible Mechanisms for the Formation of Gender Specificities in Cardiac Troponins

Under physiological conditions, the most common causes of elevated cardiac troponins are physical activity and psycho-emotional stress [65,66,67,68,69]. These physiological conditions can lead to myocardial overload, small-scale processes of cardiomyocyte apoptosis due to increased activity of the sympathoadrenal system, increased activity of pro-oxidant mechanisms, reversible damage to cardiomyocyte cell membranes, which is accompanied by the release of cytosolic troponin molecules, and a slight increase in serum concentrations of cardiac troponins [70,71,72]. Thus, elevated levels of troponins in healthy individuals may reflect the response of myocardial cells to the influence of stress factors. However, in men and women, the activity of protective mechanisms of different cells, including the myocardium, against stress factors differs, which may be a possible explanation for gender differences in serum levels of cardiac troponins. Thus, a recent study [73] demonstrated that the levels of cardiac troponin T after the same physical activity in male athletes were significantly higher than in female athletes, which is indicative of the different response of cardiac myocytes to physical activity in men and women.

In addition to this, the study by N. Tiller et al. also showed more pronounced disorders of the physiology of the cardiovascular system in men than in women after an ultramarathon [74]. Potentially, these negative effects could lead to a greater release of cardiac troponins in men than in women.

The fact that men are less protected from myocardial injury was also demonstrated by the study which observed that, after heart surgery, men had a greater increase in blood serum troponin levels than women [75].

That said, the groups of men and women were formed in accordance with the same characteristics (same body mass index, duration of artificial circulation, duration of aortic compression during surgery, etc.), which could potentially affect the degree of myocardial damage and the release of cardiac troponins. Thus, variations in damage and release of cardiac troponins are apparently due to the gender-based differences in the degree of ischemia–reperfusion injury of cardiomyocytes.

Gender differences in the degree of myocardial damage can be explained by gender specificities in the levels of a number of biologically active molecules, and in particular sex steroids. Thus, in women, estrogen levels are significantly higher than in men, in whom the predominant steroid is testosterone. That said, estrogens, unlike testosterone, are characterized by numerous cardioprotective effects. Thus, it has been shown that estrogens can have a protective effect against oxidative damage to cells by reducing the oxidative damage and stimulating the expression of antioxidant enzymes [76,77]. In addition, estrogens increase the expression of endothelial nitric oxide synthase, which leads to an increase in the formation of one of the most powerful vasodilators—nitric oxide. his, in turn, contributes to greater resistance in the cardiovascular system to coronary vessel spasms (and, accordingly, to a decrease in myocardial blood content), occurring against a background of psycho-emotional stress. Since estrogen production decreases with age in women, cardioprotective effects also decrease, which are expressed by higher levels of cardiac troponins in elderly women [56], as reported above in the previous section. Thus, the cardioprotective effects of estrogens can neutralize the degree of damage to cardiomyocytes both under physiological conditions (under stress conditions) and in case of pathological conditions.

Metabolism and renewal of cardiac myocytes [18,78,79,80,81,82,83], which is responsible for the formation of basic serum levels of cardiac troponins, is regarded as another physiological mechanism for the release of cardiac troponins. Taking into account the fact that myocardial hypertrophy is associated with cardiac troponin levels in healthy individuals [13,84,85,86,87,88], and in men the myocardial mass (physiological hypertrophy) is bigger than in women [56,61], the metabolism and renewal of cardiomyocytes can also be considered as a possible mechanism that explains the gender-based variations in the serum levels of cardiac troponins. This mechanism can also explain the gender-based differences in CK and CK–MB levels demonstrated in clinical studies [89,90,91,92,93,94,95].

## 5. Conclusions

Thus, the introduction of highly sensitive immunological tests for determining troponins into clinical practice requires the consideration of a number of biological factors in individuals, including gender and age-related characteristics. The optimal level of the 99th percentile is of great importance for the timely diagnosis of ACS, and at the same time prevents overdiagnosis of myocardial infarction. Thus, a number of studies have shown that the use of the common 99th percentile can lead to underdiagnosis of ACS in women, since their physiological levels of cardiac troponins are much lower. At the same time, the use of the 99th percentile without taking into account the gender factor leads to overdiagnosis of ACS in men, which is due to higher physiological levels of cardiac troponins in the blood. According to the IFCC, gender specificities of the 99th percentile are characteristic of most of the existing highly sensitive troponin laboratory tests. Possible mechanisms underlying gender-based variations in cardiac troponin levels are the effects of sex hormones and differences in the myocardial mass. Thus, estrogens have cardioprotective effects, as a result of which they cause expansion of the coronary vessels, which increases the resistance of cardiomyocytes to physical activities and stressful situations. In addition to the above, estrogens reduce oxidative stress, and thanks to this, the damage to cell membranes is limited and the apoptotic mechanisms are suppressed. Apparently, the complex cardioprotective effects of estrogens limit the release of cytoplasmic cardiac troponin molecules from the cardiomyocyte into the bloodstream. Further research is needed on the gender specificities of cardiac troponin levels, both of a clinical (to clarify their significance in the algorithms for diagnosing ACS) and fundamental nature (to clarify the molecular mechanisms underlying the gender-based variations in the “serum” troponin levels).

## Figures and Tables

**Figure 1 life-13-00267-f001:**
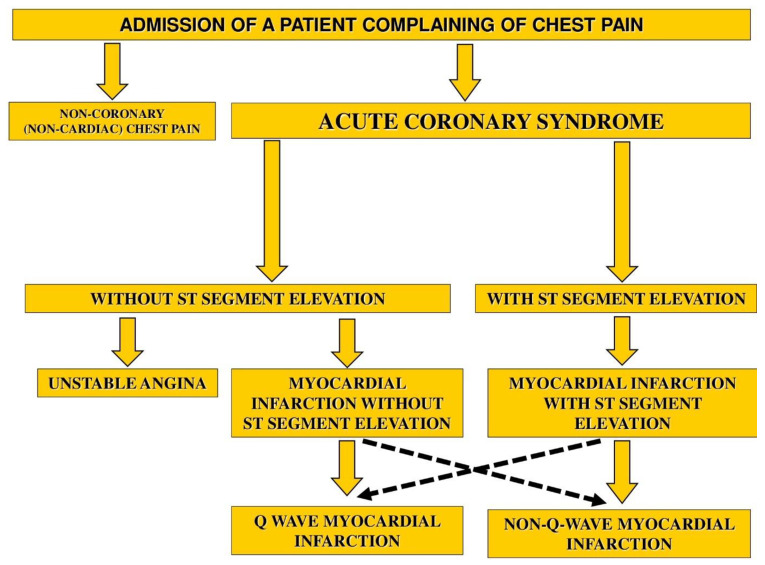
Transformation of ACS after additional examination.

**Table 1 life-13-00267-t001:** The main causes of myocardial damage (physiological and pathological) and increased concentration of cardiac troponins.

The Nature of Myocardial Damage	Nosologies/Reasons
Myocardial damage caused by ischemia	Myocardial infarction
Myocardial damage not caused by ischemia in cardiac pathologies	- Myocarditis, endocarditis, pericarditis; - The use of cardiotoxic drugs, for example, anthracyclines; - Radiofrequency or cryoablation therapy; - Cardiomyopathy and heart failure; - Pacing or defibrillation; - Infiltrative pathologies of the heart, for example, amyloidosis; - Takotsubo syndrome
Myocardial damage not caused by ischemia in extracardiac pathologies	- Sepsis - Chronic renal failure; - Chronic obstructive pulmonary disease; - Pulmonary embolism; - Neurogenic diseases (stroke, subarachnoid hemorrhages); - Arterial hypertension
Physiological conditions	- Physical activity; - Stressful situations
Without myocardial damage (false positive factors)	- Heterophilic antibodies; - Alkaline phosphatase; - Hemolysis of the sample; - Rheumatoid factor; - Fibrin clots in the sample; - Cross-reaction of diagnostic antibodies with skeletal troponins

**Table 2 life-13-00267-t002:** The main analytical characteristics, including gender specifications, of modern highly sensitive troponin immunotests [32].

Manufacturer of the Immunotest	LoD, ng/L	CV, %	Characteristics of the Examined Healthy Population, Number, Gender, Age	99 Percentile, ng/L	Type of Test Sample	Percentage of Measurable Values (Above LoD, But Less Than 99 Percentile), %
Abbott/Alinity i systems/Alinity i STAT High Sensitive Troponin-I; commercial-OUS	1.6	4.0	*n* (21–75 years old) = 1530, including M (21–75 years) = 764, W (21–75 years) = 766	Total = 26.2. For M = 34.2. For W = 15.6	Plasma with anticoagulants (lithium heparin or K2EDTA, or K3EDTA) or Serum	Total = 85, for M = 92. for W = 78
Beckman Coulter/Access 2, DxI/Access hsTnI; commercial—OUS	1.0–2.3	3.7	*n* (21–99 years old) = 1089, including M = 595, W = 494	Total = 17.5. For M = 19.8. For W = 11.6.	Plasma containing lithium heparin	Total > 50, no data on gender characteristics
bioMérieux VIDAS High Sensitive Troponin I; commercial	3.2	7.0	*n* (41–80 years old) = 815, including M (41–80 years old) = 447, W (41–80 years old) = 368.	Total = 19.0. For M = 25.0. For W = 11.0.	Serum or Plasma with anticoagulant (lithium heparin)	No data provided
ET Healthcare Pylon hsTnI assay; China FDA approved	1.2–1.4	10.0	*n* (15–91 years old) = 863, including M = 425, W = 438.	Total = 27.0. For M = 27.0. For W = 21.0.	Plasma with anticoagulant (EDTA) or serum or whole blood	Total = 91, for M = 94, for W = 89
Fujirebio Lumipulse G G1200 and G600II hsTnI	2.1	≤4.6	*n* (18–90 years old) = 1018, including M = 590, W = 428.	Total = 29.6%, For M = 32.8. For W = 27.8.	Plasma with anticoagulant (lithium heparin)	Total = 65, no data on gender characteristics
LSI Medience (formerly Mitsubishi) PATHFAST cTnI; commercial	1.0	<6.0	*n* (18–86 years old) = 474, including M = 238, W = 236	Total = 15.48. For M = 16.91. For W = 11.46	Plasma with anticoagulant (sodium heparin or lithium heparin or EDTA)	Total = 76.3, no data on gender characteristics
Ortho/VITROS/hsTroponin I; commercial	0.39–0.86	<10.0	*n* (22–91 years) = 952, including M = 466, W = 486	Total = 11.0. For M = 12.0. For W = 9.0.	Blood serum	Total > 50, no data on gender characteristics
Quidel/Alere Triage True hs-cTnI	0.7–1.6	5.0–5.9	*n* = 789, including M = 398, W = 391	Total = 20.5. For M = 25.7. For W = 14.4	Blood plasma with anticoagulant EDTA	Total ≥ 50, no data on gender characteristics
Roche/cobas e601, e602, E170/cTnT-hs 18-min; commercial	2.05	<10	*n* (20–71 years old) = 533, including W = 49.7%	Total = 14.0. For M = 16.0. For W = 9	Serum or plasma with anticoagulants (EDTA, heparin)	Total > 71.5, no data on gender characteristics
Siemens ATELLICA High-Sensitivity TnI (TnIH), US and OUS; commercial	1.6	<4.0	*n* (22–91 years old) = 2007, including M = 1000, W = 1007.	Total = 45.2. For M = 53.5. For W = 34.1.	Serum or plasma with anticoagulant (heparin)	Total = 71, for M = 84, for W = 58.

Note. LoD is the detection limit (minimum detectable concentration), CV is the coefficient of variation. M—men. W—women, EDTA—ethylenediaminetetraacetate. A more complete version of the table for other test systems can be found on the IFSC website [32].

## Data Availability

Not applicable.

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
