# Peer review of "Gender Specificities of Cardiac Troponin Serum Levels: From Formation Mechanisms to the Diagnostic Role in Case of Acute Coronary Syndrome"

_life, 2023, doi:10.3390/life13020267_

Round 1

Reviewer 1 Report

Author reviewed gender specificities of cardiac troponin levels, and it is interesting review.

I have minor comment.

Reference 14 and 15 need more introduction because readers might be curious what is gender-based approaches in ACS.

In figure 1, NON-CORONARY CHEST PAIN should be before Acute coronary syndrome.

Author Response

Dear Reviewer,

Thank You very much for reviewing the manuscript and valuable comments. In accordance with Your recommendations, I have added information about a gender-oriented approach to early diagnosis of ACS (please see page 3) and corrected Figure 1 (please see page 4). Corrections are highlighted in green.

Best Regards, 05.11.2022

Reviewer 2 Report

Comprehensive and well written review.

Substitute use of 'immunochemical' with 'immunological' tests  throughout text. 

Line: 359. Did the authors mean increased resistance or increased tolerance? due to Estrogens.

Author Response

Dear Reviewer,

Thank You very much for reviewing the manuscript and valuable comments.

In accordance with Your recommendations, I have replaced the term "immunochemical" (highlighted in green). I meant "increased resistance" due to estrogens (highlighted in green). In my opinion, in this context, "resistance" and "tolerance" are synonymous.

Best Regards, 05.11.2022

Reviewer 3 Report

It is a very interesting review where gender is considered as a determinant to consider troponin cutoff points. The gender-specific 99th percentile was characteristic of some moderately sensitive test systems, in which troponin levels were 1.3 to 5 times higher in men than in women Saenger et al. showed that statistically significant differences were observed in High-sensitivity troponin T concentrations in men and women, with the 99th percentile cutoff for men (15.5 ng/L) approximately 1.7 times higher than for women (9.0 ng/L) The threshold level of cardiac troponin in women is critical, since an incorrect decision can lead to an incorrect interpretation of the result and subsequent management of these patients. Item 4 points out possible mechanisms for the formation of gender specificities of cardiac troponins- Highly sensitive immunochemical tests for determining troponins in clinical practice require consideration of a number of biological factors of individuals, including age and gender-related characteristics. It is a complete review where the different cut-off points or 99% percentile value in men and women are ranked. The optimal level of the 99th percentile is of great importance for the timely diagnosis of acute coronary syndrome and at the same time prevents overdiagnosis of myocardial infarction.  

Author Response

Dear Reviewer,

Thank You very much for reviewing the manuscript.

Best Regards, 05.11.2022